# Effects of Elastic Resistance Exercise on Postoperative Outcomes Linked to the ICF Core Sets for Osteoarthritis after Total Knee Replacement in Overweight and Obese Older Women with Sarcopenia Risk: A Randomized Controlled Trial

**DOI:** 10.3390/jcm9072194

**Published:** 2020-07-11

**Authors:** Chun-De Liao, Yen-Shuo Chiu, Jan-Wen Ku, Shih-Wei Huang, Tsan-Hon Liou

**Affiliations:** 1Department of Physical Medicine and Rehabilitation, Shuang Ho Hospital, Taipei Medical University, New Taipei City 23561, Taiwan; 08415@s.tmu.edu.tw (C.-D.L.); 13001@s.tmu.edu.tw (S.-W.H.); 2Department of Orthopedics, Shuang Ho Hospital, Taipei Medical University, New Taipei City 23561, Taiwan; 08441@s.tmu.edu.tw; 3Department of Radiology, Shuang Ho Hospital, Taipei Medical University, New Taipei City 23561, Taiwan; 09316@s.tmu.edu.tw; 4Graduate Institute of Sports Science, National Taiwan Sport University, Taoyuan 33301, Taiwan; 5Department of Physical Medicine and Rehabilitation, School of Medicine, College of Medicine, Taipei Medical University, Taipei 11031, Taiwan

**Keywords:** osteoarthritis, sarcopenia, obesity, elastic resistance training, knee arthroplasty, ICF

## Abstract

(1) Background: Knee osteoarthritis (KOA) and aging are associated with high sarcopenia risk; sarcopenia may further affect outcomes after total knee replacement (TKR). Elastic resistance exercise training (RET) limits muscle attenuation in older adults. We aimed to identify the effects of post-TKR elastic RET on lean mass (LM) and functional outcomes in overweight and obese older women with KOA by using the brief International Classification of Functioning, Disability and Health Core Set for osteoarthritis (Brief-ICF-OA). (2) Methods: Eligible women aged ≥60 years who had received unilateral primary TKR were randomly divided into an experimental group (EG), which received postoperative RET twice weekly for 12 weeks, and a control group (CG), which received standard care. The primary and secondary outcome measures were LM and physical capacity, respectively, and were linked to the Brief-ICF-OA. The assessment time points were 2 weeks prior to surgery (T_0_) and postoperative at 1 month (T_1_; before RET) and 4 months (T_2_; upon completion of RET) of follow-up. An independent t test with an intention-to-treat analysis was conducted to determine the between-group differences in changes of outcome measures at T_1_ and T_2_ from T_0_. (3) Results: Forty patients (age: 70.9 ± 7.3 years) were randomly assigned to the EG (*n* = 20) or CG (*n* = 20). At T_2_, the EG exhibited significantly greater improvements in leg LM (mean difference (MD) = 0.86 kg, *p* = 0.004) and gait speed (MD = 0.26 m/s, *p* = 0.005) compared with the CG. Furthermore, the EG generally obtained significantly higher odds ratios than the CG for treatment success for most Brief-ICF-OA categories (all *p* < 0.001). Conclusions: Early intervention of elastic RET after TKR yielded positive postoperative outcomes based on the Brief-ICF-OA. The findings of this study may facilitate clinical decision-making regarding the optimal post-TKR rehabilitation strategy for older women with KOA.

## 1. Introduction

Patients with knee osteoarthritis (KOA) experience mobility impairment and limited function in their daily activities because of knee pain [1]. Exercise therapy may exert benefits for mild to moderate KOA, especially the obese older adults [2], and total knee replacement (TKR) surgery is an option for patients who experience severe pain, exhibit obvious radiographic evidence of KOA, and have responded poorly to conservative management [3]; TKR can relieve symptoms of KOA such as knee pain and stiffness. The progress of osteoarthritis has been attributed to aging-related decline in muscle mass (i.e., sarcopenia) [4,5,6]. Toda (2010) indicated that older women with KOA have a markedly lower percentage of lean body mass compared with their peers without KOA [6]; Lee (2016) observed that low skeletal muscle mass index in the legs is an independent risk factor for KOA [7]. Similarly, Kim reported that skeletal muscle mass index is significantly negatively associated with the Kellgren and Lawrence grade of KOA. Therefore, older people with KOA are considered at high risk of sarcopenia [5,6].

Aging-related loss of skeletal muscle mass is more obvious in the elderly with chronic disorders, especially for those who recently experienced a hospital stay for few days or weeks due to a surgery such as TKR [8]. In addition, such muscle attenuation is primarily characterized by type II myofiber phenotype atrophy and has accounted for smaller muscle fibers rather than a reduction in fiber number [9]. Previous authors have reported that resistance exercise training (RET) alleviates aging-related type II myofiber phenotype atrophy by promoting satellite cell proliferation and increasing the rate of muscle contractile and mitochondrial protein synthesis, which further contribute to myofiber hypertrophy [9]. In line with the recommended clinical practice guidelines for sarcopenia [10], RET interventions may also exert benefits for older individuals with KOA to prevent sarcopenia. However, the effects of a RET on lean mass (LM) are inconsistent among patients with KOA [11,12,13]. Some authors identified significant increases in LM responding to a RET [13] in obese patients with KOA whereas others observed no changes in LM after RET interventions [11,12]. It is vital to identify the efficacy of a RET on LM gains for KOA since patients with KOA have faced potential sarcopenia risks.

The previous studies have supported the benefits of elastic RET to prevent muscle loss in older adults [14,15]. In addition, in an RET intervention using TheraBand elastic strips or tubing in a home-based exercise program, the patients demonstrated high adherence to the program [16]. By contrast, few studies have investigated the effects of elastic RET on postoperative outcomes in older women who have received TKR [17]. Because RET is recommended in the practice guidelines for postoperative rehabilitation [18,19], it is important to identify the effect of elastic RET on muscle mass gains and function recovery after TKR.

The International Classification of Functioning, Disability and Health (ICF) is a suitable framework for the comprehensive assessment of health and functioning of an individual [20]. The ICF classification system provides a uniform and standardized language for the description of health and health-related status, based on which the ICF Core Set for osteoarthritis (ICF-OA) were developed to facilitate the application of the ICF framework in defining the typical spectrum of functional problems in patients with KOA [21]. Within the ICF-OA model, KOA may affect functioning in three ways, impairments in body functions and structures, limitations of activity levels, and participation restrictions, which are useful indicators for assessing postoperative outcomes during follow-up after TKR [22].

The primary objective of the present study was to identify the effects of a post-TKR elastic RET intervention on muscle mass in older overweight and obese women with KOA; the secondary objective was to identify changes in the qualifier of the brief ICF-Osteoarthritis (Brief-ICF-OA) categories linked with postoperative outcome measures.

## 2. Methods

### 2.1. Ethics Approval and Study Design

The present study was a randomized controlled trial with a prospective design of two parallel study arms. This study was conducted at the rehabilitation department of a university-based hospital (Taipei Medical University-Shuang Ho Hospital); ethical approval was provided by the Joint Institutional Review Board of Taipei Medical University (trial number: N201605007). In addition, this study was registered at the Chinese Clinical Trial Registry (registry number: ChiCTR-IPR-17012106) on the basis of the approved protocol. Because an effect of sex on body composition was identified during an investigation on LM outcomes in an elderly population with KOA [23], and because obesity and female sex predict poor post-TKR functional outcomes [24,25], we designed a sex-specific intervention tailored for overweight and obese older women. All patients were enrolled from August 2017 to March 2018. All the patients provided consents at the baseline admission 2 weeks prior to surgery (*T*_0_). After included, all the patients’ demographic data and prevalent comorbidities were assessed by a standard medical chart review and a comorbidity score for each patient was calculated using the comorbidity illness rating scale [26]. After discharge from the inpatient department, each enrolled patient was randomly assigned to an experimental group (EG) or control group (CG) through a randomization method (described in Section 2.4). Two examiners who were blinded to the group assignment collected physical and functional outcome data at the following time points: 2 weeks prior to surgery (*T*_0_), 1 month of follow-up postoperatively (before elastic RET intervention; *T*_1_), and 4 months of follow-up postoperatively (upon completion of elastic RET intervention; *T*_2_). Muscle mass was assessed at *T*_0_ and *T*_2_. All collected data of outcome measures were mapped and linked to the Brief-ICF-OA through use of the ICF linking rules [27] by two senior medical doctors (THL and SWH) familiar with the ICF [28].

### 2.2. Participants

Women aged 60–85 years who were overweight or obese (body mass index ≥ 24 kg/m^2^), had potential sarcopenia risk (usual gait speed < 0.8 m/s [29]), had received radiographic diagnoses of KOA (Kellgren and Lawrence grade ≥ 3), and were scheduled to receive primary TKR were recruited from the outpatient clinic of the rehabilitation department at Taipei Medical University-Shuang Ho Hospital. The patients were excluded if they had uncontrolled hypertension, any cardiovascular or pulmonary disease that would prevent them from engaging in an exercise study, or neurological or cognitive impairment.

### 2.3. Sample Size Estimation

A primary outcome in this study was appendicular LM gain. According to the results of our previous study, which compared the effects of elastic RET with those of no RET in obese older women, the effect size of the proposed elastic exercise training regimen was estimated to be approximately 0.9 for appendicular LM [15]. To identify a significant between-group difference of 0.99 kg in appendicular LM, we estimated (on the basis of previous results [15]) that 32 patients would be required to achieve 80% power and an alpha value of 0.05 when a standard deviation (SD) of 1.10 kg was assumed [30]. After anticipating a dropout rate of 20%, we enrolled 40 patients.

### 2.4. Randomization

All the patients were randomly assigned to either the EG or CG at *T*_1_ using sealed envelopes. Group assignment was carried out independently for both groups by an independent principal investigator who was not involved in the enrollment, intervention, or assessment. A list of computer-generated random numbers provided by an independent randomization center was used in stratified permuted blocks of size four.

### 2.5. Interventions

#### 2.5.1. Inpatient Rehabilitation

Immediately after TKR, all patients were entered to engage in a standardized rehabilitation program during their hospital stay; the program included continuous passive motion exercise and training for functional reconditioning.

#### 2.5.2. Elastic Resistance Exercise

The exercise intervention began 2 weeks after inpatient discharge. We conducted a home-based RET program based on our previously reported exercise protocol [15]; the program required TheraBand (Hygenic Co., Akron, OH, USA) products, whose colors indicate the resistance level (yellow, red, green, blue, black, or gray). The first 2 weeks of elastic RET sessions served as a familiarization period to ensure that each exercise was performed using the correct technique both at home and at the clinic. During the familiarization period, all patients in the EG were trained and supervised by a licensed senior physical therapist who was blinded to the group assignment; after the familiarization period, the RET protocol was continued through home-based training sessions.

The RET protocol followed the guidelines for resistance training for older individuals provided by the American College of Sports Medicine [31]. Each patient in the EG underwent two training sessions per week for 12 weeks, yielding a total of 24 sessions within 3 months. Each training session comprised a 10-min warm-up, 40-min period of elastic resistance exercises, and 5-min cool-down period. The major systemic muscle groups of the upper and lower quarter were targeted, and the following seven exercises were incorporated into the training design: seated chest press, seated row, seated shoulder press, knee extension, knee flexion, hip flexion, and hip extension (Appendix A). Progressive resistance load training was used to increase the difficulty of the exercise (by changing band color) every 2 weeks (Appendix A). Participants in the EG performed three sets of 10–20 repetitions for each movement. The exercise loads with respect to individual yielding elasticity (as indicated by band color) in the resistance training were set at levels that the patients perceived as *somewhat difficult* (grade 13) to *difficult* (grade 15) according to the 15-point Borg Scale of Perceived Exertion [32], representing exercise of moderate intensity (65%–80%, maximum of one repetition) [33].

All patients in the EG received a logbook containing detailed instructions for the prescribed exercises and were asked to record the number of repetitions completed in each RET session. The physical therapist contacted the patients every 2 weeks by telephone and modified their target numbers of repetitions as necessary. During the 12-week RET intervention, the patients attended 22 ± 1.6 exercise sessions and had a mean attendance rate of 91.6%.

#### 2.5.3. Standard Care

The CG received standard care, including KOA education, pharmacological therapy, and physical therapy, none of which were involved related to muscle strengthening exercises but rather focused on functional training such as stretching exercises, active range of motion (ROM) exercises, stationary cycling, and treadmill walking. These patients were instructed to maintain their preoperative levels of physical activity after TKR.

### 2.6. Outcome Measures

#### 2.6.1. Primary Outcome Measures

The primary outcomes of interest in this study included body composition and muscle mass indices which have been used as indicators of sarcopenia [29]. Skeletal muscle mass, including arm and leg LM, was measured using a Hologic QDR-1000/W whole-body dual-energy X-ray absorptiometer (Hologic, Waltham, MA, USA) at *T*_0_ and *T*_2_. All scans and analyses were conducted by the same investigator, who was blinded to the patients’ group allocations to minimize interobserver variation. The absorptiometers were used previously to identify body composition after TKR [34]. To minimize bias when estimating the LM of the affected leg, we adopted half-body scanning, through which the LM of the nonoperated leg was used to estimate that of the operated leg [35]. Appendicular LM was calculated as the sum of arm and leg LM, which was then converted into the appendicular LM index (AMI, kg/m^2^) by dividing appendicular LM by squared body height [29].

#### 2.6.2. Secondary Outcome Measures

*Active knee flexion ROM*. The joint flexibility of interest in this study was active knee flexion ROM because preoperative and postoperative knee flexion are significantly associated with daily activity following TKR [36,37]. Active knee flexion ROM is defined as the maximal extent to which a patient can actively bend the tested knee [38].

*Physical capacity*. Physical capacity was assessed using functional mobility tasks, including gait speed (m/s) and timed chair rise (TCR; repetition) tests that were applied in our previous study to assess outcomes after TKR [39].

*Western Ontario and McMaster Universities Osteoarthritis Index*. The Western Ontario and McMaster Universities Osteoarthritis Index (WOMAC) assesses functional outcomes after TKR in patients with KOA; the Chinese version of the WOMAC questionnaire was used in this study [40]. The WOMAC questionnaire consists of three domains containing a total of 24 items, with 5, 2, and 17 items assessing pain, stiffness, and physical function, respectively. The dimension scores for the WOMAC index range from 0 to 100, with 100 indicating the least healthy state possible.

*Brief-ICF-OA.* The ICF-Osteoarthritis set consists of 55-s-level categories selected from the entire ICF classification system of 362 categories [21]. We used the Brief-ICF-OA, which comprises three categories each for bodily functions, bodily structures, and activities and participation, and four categories for each of these component’s environmental factors [21]. A qualifier scale was used to rate each category (responses from 0 to 4: 0 = unaffected; 1 = mildly affected; 2 = moderately affected; 3 = severely affected; and 4 = completely disabled) to assess the extent of the patients’ problems with respect to the bodily functions, bodily structures, and activities and participation components [20].

### 2.7. Statistical Analysis

An intention-to-treat analysis was conducted to minimize bias related to the loss of follow-up data, and the last-observation-carried-forward technique was used to impute any missing data [41]. The Kolmogorov–Smirnov test was performed to confirm the normality of the distributions of all outcome variables for both groups. Independent *t* tests were performed to determine the between-group differences in changes of outcome measures at *T*_1_ and *T*_2_ from *T*_0_.

The tested values of all outcome measures at each time point were mapped to the first qualifier in the linked ICF code. The following cut-off points for arm and leg LM, AMI, ROM, and mobility scores were used to classify patients as unaffected (qualifier 0): ≥3.55 kg and ≥13.5 kg for arm and leg LM, respectively [42]; ≥6.12 kg/m^2^ for AMI [29]; ≥130° for knee ROM [43]; ≥1.1 m/s for gait speed [44]; and ≥14 repetitions for the TCR test [45]. The means of the qualifiers for all Brief-ICF-OA categories at each time point and the mean changes at *T*_2_ from *T*_0_ and *T*_1_ were calculated and analyzed. In addition, the treatment success rate was determined as the percentage of patients with qualifier values of 0 (unaffected) for all Brief-ICF-OA categories; furthermore, the treatment success rates of the groups were compared with respect to each linked Brief-ICF-OA code, and the result of each comparison was presented as an odds ratio with a 95% confidence interval (CI).

Bonferroni correction was employed to control the familywise error rate following multiple comparisons [46]. Considering that eight comparisons were performed, in accordance with the number of linked Brief-ICF-OA categories in this study, comparisons with *p* < 0.006 were considered statistically significant differences and are presented as means and SDs in this study. SPSS version 22.0 was used for all analyses.

## 3. Results

### 3.1. Patient Demographics and Clinical Characteristics

A total of 47 patients were recruited for this study (Figure 1); among those, 7 were excluded, and the other 40 were enrolled and randomly allocated to the EG (*n* = 20) or CG (*n* = 20). Eighteen patients in the EG and 17 patients in the CG completed the 4-month follow-up assessment at *T*_2_. The patients’ demographics and clinical characteristics are presented in Table 1.

### 3.2. Effects of Elastic RET on Muscle Mass

All outcome variables for both groups were confirmed to be normally distributed and were mapped according to each category in the Brief-ICF-OA, as detailed in Appendix A. A total of 8 of the 13 categories in the Brief-ICF-OA were identified in this study; the primary outcome measures were linked with codes s730 (arm LM), s750 (leg LM), and s799 (AMI) in the bodily structures component.

Table 2 presents the baseline scores and mean changes at *T*_1_ and *T*_2_ for all muscle mass outcomes. The 12-week elastic RET intervention achieved significantly greater changes in leg LM (mean difference (MD) = 0.86 kg, 95% CI = 0.29–1.42, *p* = 0.004) at *T*_2_ compared with the CG results.

### 3.3. Effects on Pain, ROM, Physical Capacity, and WOMAC Outcomes

The baseline scores and mean changes at *T*_1_ and *T*_2_ for all secondary outcomes that were strongly linked to Brief-ICF-OA codes are presented in Table 2. The patients in the EG exhibited significantly greater changes at *T*_2_ in the TCR task (MD = 6.8 repetitions, 95% CI = 2.77–10.83, *p* = 0.002) and walking speed (MD = 0.26 m/s, 95% CI = 0.08–0.43, *p* = 0.005) compared with the CG, corresponding to ICF codes b730 (muscle power functions) in the bodily functions component and d540 (walking) in the activities and participation component, respectively.

### 3.4. Results of Function and Disability Based on the Brief-ICF-OA

Table 3 shows the mean changes in the qualifier of each ICF code at *T*_2_ and *T*_1_ from *T*_0_. After 12 weeks of elastic RET (*T*_2_), the EG exhibited significantly greater improvement in the qualifier of ICF code b730 (muscle power functions) in the bodily functions component compared with the CG (MD = −1.5, *p* = 0.001). In the activities and participation component, significantly greater improvements were observed in the EG for codes d450 (walking; MD = −1.1, *p* = 0.002) and d540 (dressing; MD = −1.1, *p* = 0.004) compared with the CG.

Figure 2 presents the effects of elastic RET on the treatment success rate. After 12 weeks of elastic RET (*T*_2_), the EG exhibited significantly higher odds ratios for treatment success than the CG for most linked categories of the Brief-ICF-OA; however, the percentages of patients who yielded a qualifier value of 0 for pain (code b280) did not differ significantly between the two groups.

## 4. Discussion

We investigated the effects of 12-week elastic RET using TheraBand products on postoperative outcomes using the Brief-ICF-OA among patients who had received primary TKR surgery. Compared with the CG, the EG exhibited significant improvements in leg LM, mobility measures, and the qualifiers of the mapped ICF components of Bodily functions, Bodily structures, and Activities and participation. In addition, the EG obtained higher treatment success rates than did CG for most Brief-ICF-OA categories.

Osteoarthritis reportedly reduces muscle mass [4,5,6] and thus hinders the efficiency of muscle function [47]. Furthermore, muscle mass deficit may persist even after TKR surgery. In the present study, 50% to 55% of the patients exhibited low muscle mass prior to surgery; after 4 months of follow-up after TKR (*T*_2_), the patients who had received elastic RET (the EG) exhibited significant improvement in leg LM, whereas those who received functional training (the CG) experienced additional muscle attenuation. In addition, analysis of the Brief-ICF-OA categories in the bodily structures component associated with muscle mass revealed that the EG had a higher treatment success rate than the CG in terms of prevention of sarcopenia (Figure 2); among the 20 patients in the EG, 13, 10, and 17 obtained qualifier values of 0 for arm LM (code s730), leg LM (code s750), and AMI (code s799), respectively. By contrast, in the CG, only 6, 2, and 7 of the 20 patients obtained qualifier values of 0 for these codes, respectively. Our results indicated that RET can be prescribed to preserve muscle mass after TKR surgery; these results are consistent with those of a previous study, which recommended RET for older women with sarcopenia [15].

Previous studies have indicated that older individuals with sarcopenia who receive RET interventions with training durations of 8–24 weeks can achieve greater changes in appendicular LM (MD = 0.45–1.20 kg) [15,48,49,50] and AMI (MD = 0.17–1.16 kg/m^2^) [48,49,50] compared with their untrained peers, regardless of the RET protocol. The training period required to achieve evident LM gains in response to RET was estimated as 8–12 weeks for older individuals [51]. Following previous results, a 12-week RET program was employed in the present study for overweight and obese older women who had received TKR; the EG made greater gains in appendicular LM (MD = 1.09 kg) and AMI (MD = 0.24 kg/m^2^) than did the CG.

The following minimal clinically important differences in mobility outcome measures have been established for older people with KOA: 0.12 m/s for gait speed [52] and 2.6 repetitions for the TCR task [53]. The EG group in present study achieved clinically meaningful changes in walking speed (MD = 0.26 m/s) and TCR task performance (MD = 6.8 repetitions); in addition, an increase of 7.8% in leg LM was observed in the EG. These changes in muscle mass and physical mobility are synergistic, as supported by a previous study that indicated that appendicular LM gains significantly predict positive outcomes for walking capability and leg strength in response to exercise interventions [54].

At *T*_2_, the EG exhibited greater recovery from preoperative baseline (*T*_0_) than the CG in terms of most functional outcomes, excluding knee active flexion ROM. The between-group MD in knee ROM at *T*_2_ was 10.4°; this result has clinical relevance [55]. However, the between-group difference in changes at *T*_2_ from *T*_0_ was not statistically significant. Because the mean preoperative ROM in the CG (81.8°) was lower than that in the EG (87.4°), the results indicated that preoperative ROM may influence total ROM restoration after TKR surgery. The results of this study are supported by those of our previous study, which have indicated that preoperative ROM affects knee flexion after TKR surgery [56]. From the perspective of the ICF classification system, the EG exhibited a higher odds ratio for treatment success than the CG for mobility of joint function (code b710), which affects knee ROM. This finding indicated that postoperative outcomes assessed by the ICF classification system may provide results independent of confounding factors such as preoperative patient conditions.

This study had some limitations. First, our study recruited only older women and did not consider long-term outcomes. Because of the sex-specific response to RET, our results may not be generalizable to older men with KOA. Future studies are warranted to expand the population, especially considering both genders, and to investigate long-term outcomes up to 12 months after TKR. Future studies should evaluate whether this kind of exercise can delay or prevent the onset of symptoms related to the contralateral KOA. Second, the small sample size prohibited the detection of an association between improved LM and physical mobility, despite the finding that low muscle mass is associated with low physical function and severe mobility limitations [57]. Third, contextual factors such as environmental and personal factors in the Brief-ICF-OA were not assessed in this study. Environmental factors may affect the assessment of surgical outcomes because these factors tend to be crucial concerns of patients during early postoperative follow-up [58]. Finally, we did not analyze diet or nutritional supplement control during the intervention. Therefore, we were unable to draw conclusions regarding the association between nutritional supplements and changes in LM during RET. Dietary patterns and nutritional supplements such as protein supplements may induce changes in whole-body weight or muscle mass during RET [59].

In conclusion, this randomized controlled prospective study revealed that 12 weeks of post-TKR elastic RET yielded positive muscle mass and functional mobility outcomes, as mapped with the Brief-ICF-OA, in overweight and obese older women with KOA. The results of this study suggest that elastic RET should be considered as a treatment option to help patients with KOA regain muscle mass and strength after TKR surgery. The elastic RET protocol and findings of this study could facilitate clinical decision-making regarding the optimal post-TKR rehabilitation strategy for older women with KOA, particularly those with sarcopenia.

## Figures and Tables

**Figure 1 jcm-09-02194-f001:**
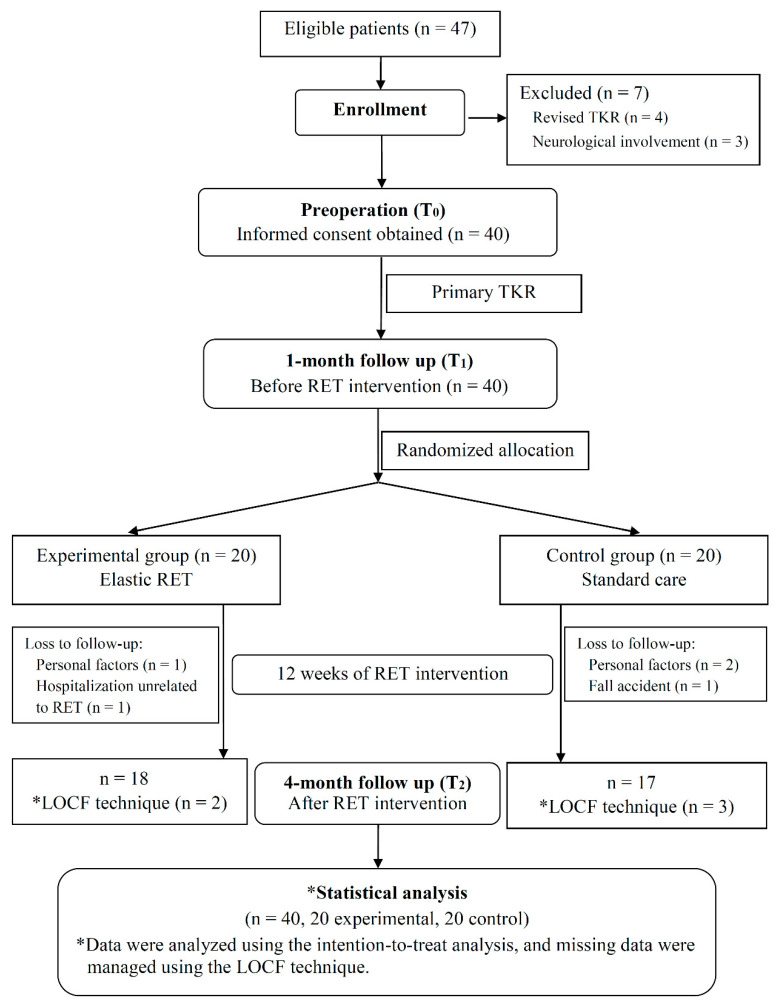
Consolidated Standards of Reporting Trials flowchart for patient enrollment and allocation in the present study. LOCF, last-observation-carried-forward; RET, resistance exercise training; TKR, total knee replacement.

**Figure 2 jcm-09-02194-f002:**
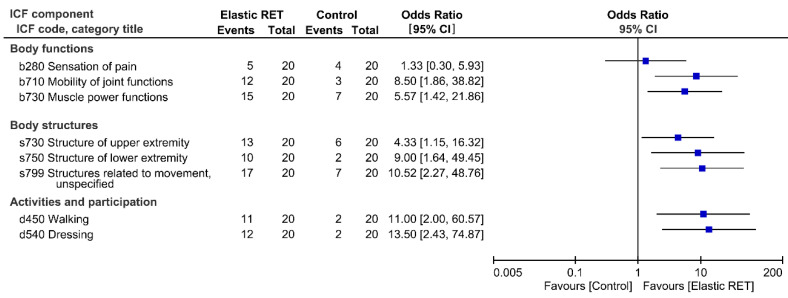
Effect on treatment success rate in all categories of the brief International Classification of Functioning, Disability and Health (ICF) Core Set for osteoarthritis. Each comparison result is represented as a point estimate (square box) with a 95% confidence interval (CI, horizontal line). Results plotted on the right side indicate positive effects of elastic resistance exercise training (RET).

**Table 1 jcm-09-02194-t001:** Patient demographics and clinical characteristics.

Variables	Experimental (*n* = 20)		Control (*n* = 20)	
	Mean	(SD)	Mean	(SD)
**Sociodemographic**				
Age (year)	72.22	(7.75)	69.79	(6.72)
Education, *n* (%)				
Primary	12	(60)	13	(65)
Secondary	4	(20)	2	(10)
University	4	(20)	5	(25)
Living status, *n* (%)				
Alone	3	(15)	4	(20)
Family/Spouse	17	(85)	16	(80)
Smokers, *n* (%)	2	(10)	3	(15)
Alcohol consumption, *n* (%)	3	(15)	2	(10)
**Clinical characteristics**				
Operated leg, Right, *n* (%)	17	(85)	15	(75)
K&L grade (Non-TKR leg), *n* (%)				
I	9	(45)	7	(35)
II	8	(40)	9	(45)
III	3	(15)	4	(20)
Use walking aids, *n* (%)	7	(35)	6	(30)
Number of comorbidities, *n* (%)				
1	8	(40)	8	(40)
2	6	(30)	5	(25)
3	3	(15)	5	(25)
≥4	3	(15)	2	(10)
Low muscle mass, *n* (%) ^a^	10	(50)	11	(55)
Acute hospital stay (day)	8.3	(1.60)	7.9	(2.10)
BMI (kg/m^2^)	28.27	(3.25)	27.60	(3.64)
AMI (kg/m^2^)	6.22	(1.10)	5.95	(0.99)
Preoperative PA				
Moderate PA (h/week)	0.97	(1.49)	1.11	(2.24)
Walk time (h/week)	6.79	(6.26)	7.11	(6.31)

^a^ Low muscle mass was identified by a cut-off value of AMI < 6.12 kg/m^2^ [29]. SD = standard deviation; BMI = body mass index; AMI = appendicular mass index; TKR = total knee replacement; K&L grade = Kellgren and Lawrence grade; PA = physical activity.

**Table 2 jcm-09-02194-t002:** Mean changes in outcome variables based on the categories of brief International Classification of Functioning, Disability and Health Core Set for osteoarthritis (Brief-ICF-OA) at time point 1 month after operation (*T*_1_) and 4 months after operation (*T*_2)_ from baseline (before operation (*T*_0_)).

ICF Component ICF Code, ICF Category Title (Measure)	Experimental (*n* = 20) *						Control (*n* = 20) *					
	*T* _0_		*T*_1_−*T*_0_		*T*_2_−*T*_0_		*T* _0_		*T*_1_−*T*_0_		*T*_2_−*T*_0_	
	Mean	(SD)	Mean	(SD)	Mean	(SD)	Mean	(SD)	Mean	(SD)	Mean	(SD)
Body functions												
b280 Sensation of pain (WOMAC−Pain)	12.50	(3.23)	−9.49	(3.09)	−6.95	(2.55)	10.28	±3.32	−8.60	(1.96)	−5.55	(1.23)
b710 Mobility of joint functions (ROM, degree)	87.37	(20.14)	4.25	(4.39)	28.08	(12.4)	81.80	±21.16	7.41	(6.64)	20.58	(11.75)
b730 Muscle power functions (TCR, repetition)	10.50	(3.69)	−1.05	(5.91)	9.05	(6.74) ^†^	11.25	±3.13	−1.80	(4.57)	2.25	(5.81)
Body structures												
s730 Structure of upper extremity (arm lean mass, kg)	3.69	(1.19)			0.24	(1.13)	3.41	±1.26			0.01	(1.02)
s750 Structure of lower extremity (leg lean mass, kg)	10.69	(1.80)			0.84	(0.91) ^†^	10.67	±1.75			−0.02	(0.89)
s799 Structures related to movement, unspecified (AMI, kg/cm^2^)	6.22	(1.10)			0.17	(0.32)	5.95	±0.99			−0.07	(0.34)
Activities and participation												
d450 Walking (gait speed, m/s)	0.75	(0.31)	−0.17	(0.31)	0.30	(0.28) ^†^	0.79	±0.37	−0.22	(0.35)	0.04	(0.27)
d540 Dressing (WOMAC−PF) ^a^	4.50	(1.82)	1.65	(0.88)	−0.95	(1.76)	3.45	±2.11	2.55	(2.44)	0.60	(2.09)

* *T*_0_, pre-operation; *T*_1_, pretest before resistance exercise training (1 month post-operation); *T*_2_, posttest at the end of exercise intervention (4 months post-operation). † A significant difference compared to the control group. ^a^ Data were calculated based on ranked score of physical function subscale of Western Ontario & McMaster Universities Osteoarthritis Index (WOMAC-PF) item 9 (putting on socks) and item 11 (taking off socks). ICF, The International Classification of Functioning, Disability and Health; Brief-ICF-OA, brief ICF Core Set for osteoarthritis; ROM, range of motion; TCR, timed chair rise; AMI, appendicular mass index; WOMAC−Pain, pain subscale of Western Ontario & McMaster Universities Osteoarthritis Index; WOMAC−PF, physical function subscale of Western Ontario & McMaster Universities Osteoarthritis Index.

**Table 3 jcm-09-02194-t003:** Mean changes in ICF-code qualifiers at time point *T*_1_ and *T*_2_ from baseline (*T*_0_).

ICF Component ICF Code, ICF Category Title	Experimental (*n* = 20)						Control (*n* = 20)					
	*T* _0_		*T*_1_−*T*_0_		*T*_2_−*T*_0_		*T* _0_		*T*_1_−*T*_0_		*T*_2_−*T*_0_	
	Mean	±SD	Mean	±SD	Mean	±SD	Mean	±SD	Mean	±SD	Mean	±SD
Body functions												
b280 Sensation of pain	3.30	(0.80)	−1.35	±0.59	−2.15	(0.88)	2.90	(0.85)	−1.20	(0.52)	−1.80	(1.06)
b710 Mobility of joint functions	1.85	(0.75)	−0.20	±0.41	−1.10	(0.64)	2.15	(0.67)	−0.50	(0.51)	−0.80	(0.41)
b730 Muscle power functions	1.60	(1.10)	0.30	±1.72	−1.75	(0.12) *	1.35	(0.99)	0.60	(1.50)	−0.25	(1.52)
Body structures												
s730 Structure of upper extremity	0.50	(0.69)			0.00	(0.56)	0.90	(0.79)			0.30	(1.03)
s750 Structure of lower extremity	1.30	(0.66)			−0.60	(0.94)	1.50	(0.61)			−0.10	(0.79)
s799 Structures related to movement, unspecified	0.40	(0.68)			−0.25	(0.85)	0.48	(1.09)			0.25	(1.16)
Activities and participation												
d450 Walking	2.00	(0.92)	0.45	±1.10	−1.75	(1.12) *	1.80	(0.89)	0.70	(0.98)	−0.25	(1.52)
d540 Dressing	2.30	(0.87)	0.35	±1.39	−1.50	(1.24) *	1.95	(1.00)	0.75	(1.37)	−0.40	(1.05)

*T*_0_, baseline (pre-operation); *T*_1_, pretest before exercise intervention (1 month post-operation); *T*_2_, posttest at the end of exercise intervention (4 months post-operation). ICF, The International Classification of Functioning, Disability and Health. * A significant difference compared to the control group.

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
