# Peer review of "Effects of Elastic Resistance Exercise on Postoperative Outcomes Linked to the ICF Core Sets for Osteoarthritis after Total Knee Replacement in Overweight and Obese Older Women with Sarcopenia Risk: A Randomized Controlled Trial"

_jcm, 2020, doi:10.3390/jcm9072194_

Round 1

Reviewer 1 Report

The aim of this study is to identify the effects of post-TKR elastic RET on lean mass (LM) and functional outcomes in overweight and obese older women with KOA by using the Brief-ICF-OA.

The article reports an innovative work. A previous study had already indagated the effects of elastic band exercise on lower limb rehabilitation of elderly patients undergoing total knee arthroplasty with a quasiexperimental design in a group composed mainly but not completely by woman (Chou et al. 2019).

The aims of the study are well explained. The structure of the study is well organized. The materials and methods sectionsareedited with precision, the methods are appropriate to the research question and they are discussed in detail so it is easy to understand the proceduresactually followed. Statistical analysis is well conducted and appropriate. Protocols on ethical treatment were followed.

The results are clearly explained in the text and there are not repetitions between text and tables. It should be interesting also to measure in the population considered muscle strength of lower limbs measured with MRC scale.

The work weakness consists in not considering long-term outcomes and participants of both sexes.It would be desirableto conduct other studies expanding the population, especially considering both genders and repeating the measurements in different follow-up periods. In future studies, it should be interesting also to evaluate if this kind of exercise can delay or prevent the onset of symptoms related to contralateral knee osteoarthritis.

The paper is well writing, however there are some minor concerns as detailed below.

In the title it should be better to specify that the women involved in the study are characterized by sarcopenia and obesity.

The abstract is well-written and highlights the aims and the results of the study. In the keywords sectionit would be preferable to add“elastic resistance training”and “obesity” to the words chosen, in order to make keywords more representative of the article content.

A study conducted by Masiero S. et al. (2018) evaluated hydrokinesitherapy effects in thermal setting in obese patients with knee osteoarthritis (Masiero, S., Vittadini, F., Ferroni, C. et al. The role of thermal balneotherapy in the treatment of obese patient with knee osteoarthritis. Int J Biometeorol 62, 243–252 (2018). https://doi.org/10.1007/s00484-017-1445-7). It may be cited in the introduction paragraph.

The article is written in good English, even if there are some little concerns:

  • Line 17: it should be better to write “ageing” instead of “age”.
  • line 61: it should be better to write “a RET” or “an elastic RET”.

Regarding the clarity of the article content, it may be appropriate to report some sentences that need to be revised in order to make the article clearer.

  • Line 52: it may be preferable to briefly explain “a short duration of hospitalization”.
  • Line 84 and line 107: it is not specified what rehabilitation centre is the setting of the study.
  • Line 104: in the participants paragraph it is not mentioned sarcopenia as a inclusion criterion.
  • Line 206:≥55 kg.
  • Line 232: BMI is better explained as Body Mass Index; AMI=appendicular mass index.
  • Table 2: In the table description it is mentioned VAS scale but it is not reported in the table.

Author Response

Comments and Suggestions for Authors

The aim of this study is to identify the effects of post-TKR elastic RET on lean mass (LM) and functional outcomes in overweight and obese older women with KOA by using the Brief-ICF-OA.

The article reports an innovative work. A previous study had already indagated the effects of elastic band exercise on lower limb rehabilitation of elderly patients undergoing total knee arthroplasty with a quasiexperimental design in a group composed mainly but not completely by woman (Chou et al. 2019).

The aims of the study are well explained. The structure of the study is well organized. The materials and methods sections reedited with precision, the methods are appropriate to the research question and they are discussed in detail so it is easy to understand the procedures actually followed. Statistical analysis is well conducted and appropriate. Protocols on ethical treatment were followed.

The results are clearly explained in the text and there are not repetitions between text and tables. It should be interesting also to measure in the population considered muscle strength of lower limbs measured with MRC scale.

The work weakness consists in not considering long-term outcomes and participants of both sexes. It would be desirable to conduct other studies expanding the population, especially considering both genders and repeating the measurements in different follow-up periods. In future studies, it should be interesting also to evaluate if this kind of exercise can delay or prevent the onset of symptoms related to contralateral knee osteoarthritis.

Response

Thank you for your comprehensive review and constructive comments regarding our manuscript. We made the work weakness as study limitations. We revised the statements in study-limitation paragraph as follows:

Lines 328–333:

“First, our study recruited only older women and did not consider long-term outcomes. Because of the sex-specific response to RET, our results may not be generalizable to older men with KOA. Future studies are warranted to expand the population, especially considering both genders, and to investigate long-term outcomes up to 12 months after TKR. In future studies, it should be also interested in evaluating whether this kind of exercise can delay or prevent the onset of symptoms related to the contralateral KOA.”

The paper is well writing, however there are some minor concerns as detailed below.

In the title it should be better to specify that the women involved in the study are characterized by sarcopenia and obesity.

Response

According to the reviewer’s comment, we revised the title as follows:

“Effects of Elastic Resistance Exercise on Postoperative Outcomes Linked to the ICF Core Sets for Osteoarthritis after Total Knee Replacement in Overweight and Obese Older Women with Sarcopenia Risk: A Randomized Controlled Trial”

The abstract is well-written and highlights the aims and the results of the study. In the keywords section it would be preferable to add “elastic resistance training” and “obesity” to the words chosen, in order to make keywords more representative of the article content.

Response

We revised the keywords section as follows:

Line 39

“Keywords: osteoarthritis; sarcopenia; obesity; elastic resistance training; knee arthroplasty; ICF”

A study conducted by Masiero S. et al. (2018) evaluated hydrokinesitherapy effects in thermal setting in obese patients with knee osteoarthritis (Masiero, S., Vittadini, F., Ferroni, C. et al. The role of thermal balneotherapy in the treatment of obese patient with knee osteoarthritis. Int J Biometeorol 62, 243–252 (2018). https://doi.org/10.1007/s00484-017-1445-7). It may be cited in the introduction paragraph.

Response

In accordance to the reviewer’s comment, we made statements and cited the study (Masiero et al., 2018) in the introduction paragraph as follows:

Lines 43–44

“Exercise therapy may exert benefits for mild to moderate KOA, especially the obese older adults [2], and……”

The article is written in good English, even if there are some little concerns:
Line 17: it should be better to write “ageing” instead of “age”.
Response
We revised the statement as follows:
Line 18: “Knee osteoarthritis (KOA) and aging are associated with……”

line 61: it should be better to write “a RET” or “an elastic RET”.
Regarding the clarity of the article content, it may be appropriate to report some sentences that need to be revised in order to make the article clearer.
Response
We revised the statement as follows:
Lines 63–64: “However, the effects of a RET on lean mass (LM) are inconsistent among patients with KOA [11-13].”

We also made statements to clarity the article content as follows:
Lines 61–67
“In line with the recommended clinical practice guidelines for sarcopenia [10], RET interventions may also exert benefits for older individuals with KOA to prevent sarcopenia. However, the effects of a RET on lean mass (LM) are inconsistent among patients with KOA [11-13]. Some authors identified significant increases in LM responding to a RET [13] in obese patients with KOA whereas others observed no changes in LM after RET interventions [11, 12]. It is vital to identify the efficacy of a RET on LM gains for KOA since patients with KOA have faced potential sarcopenia risks.”

Line 52: it may be preferable to briefly explain “a short duration of hospitalization”.
Response
We revised the statement as follows:
Lines 54–56
“Aging-related loss of skeletal muscle mass is more obvious in the elderly with chronic disorders, especially for those who recently experienced a hospital stay for few days or weeks due to a surgery such as TKR [8].”

Line 84 and line 107: it is not specified what rehabilitation centre is the setting of the study.
Response
We revised the statement as follows:
Lines 91–92
“This study was conducted at the rehabilitation department of a university-based hospital (Taipei Medical University-Shuang Ho Hospital);”

Lines 114–116
“… were recruited from the outpatient clinic of the rehabilitation department at Taipei Medical University-Shuang Ho Hospital.”

Line 104: in the participants paragraph it is not mentioned sarcopenia as a inclusion criterion.
Response
We made the statement to describe potential risk of sarcopenia as a inclusion criterion as follows:
Lines 112–113
“Women aged 60–85 years who ……, had potential sarcopenia risk (usual gait speed <0.8 m/s [29]), ……”

Line 206:≥55 kg.
Response
We revised the statement as follows:
Lines 214
“…… were used to classify patients as unaffected (qualifier 0): ≥3.55 kg and ……”

Line 232: BMI is better explained as Body Mass Index; AMI=appendicular mass index.
Response
We corrected the words in Table 1 as follows:
Line 240
“BMI = body mass index; AMI = appendicular mass index;”

Table 2: In the table description it is mentioned VAS scale but it is not reported in the table.
Response
We removed the description of VAS scale in Table 2.

Reviewer 2 Report

This is an interesting study and a well-written article. The authors have done a great effort to come up with the concept, design and the writing of their results in a clear and concise manner. The limitations are well-acknowledged fairly.

The only slight correction I would suggest is:

Line 83: randomized ‘controlled’ trial (instead of 'control')

I would like to compliment the authors for their work.

Author Response

Response

Thank you for the comment. In the revised manuscript, we revised the word as follows:

Line 90: “The present study was a randomized controlled trial with a prospective design……”